

# Exploring hydrological similarity during soil moisture recession periods using time dependent variograms

Mirko Mälicke[1], Sibylle K. Hassler[1], Markus Weiler[2], Theresa Blume[3], and Erwin Zehe[1]

[1]Institute for Water and River Basin Management, Karlsruhe Institute of Technology (KIT), Germany
[2]Hydrology, Faculty of Environment and Natural Resources, University of Freiburg, Germany
[3]GFZ German Research Centre for Geosciences, Section 5.4 Hydrology, Germany

**Correspondence:** Mirko Mälicke (mirko.maelicke@kit.edu)

**Abstract.** This study proposes a new method for identifying temporally stable spatial patterns in soil moisture. Soil moisture patterns during rainfall-driven wetting conditions and during radiation-driven drying essentially reflect different processes with specific underlying controls. Consequently we expect that these patterns exhibit different covariance structures, and their spatial analysis should be separated accordingly. More specifically we hypothesize that: (H1): An ensemble of distributed soil moisture observations will converge to a rank-stable configuration during long-term drying periods and re-organize to stable ranks after disturbances; (H2): Variograms of these soil moisture ranks converge towards a stable configuration during drying periods, reflecting the covariance of the underlying time-invariant controls. These hypotheses were tested using soil moisture measurements which were recorded within the CAOS research unit in Luxembourg. We found evidence of stable rank configurations for time spans of several weeks. During rainfall events, these stable ranks were disturbed but later reorganized into the same pre-event configuration. Coupling time-shifting variograms with a density-based clustering algorithm enabled us to identify a convergence towards stable spatial variogram configurations. Moreover, the spatial organization of soil moisture showed preferred states with distinct patterns, depending on their respective drivers. This corroborates that the proposed method can be used to disentangle spatial structure originating from rainfall patterns from those controlled by the internal terrestrial system properties. Furthermore, we conclude that during stable states variogram aggregates originating from the density-based clustering could in principle be used for interpolation purposes, as they represent a temporally stable covariance. In contrast, an interpolation can be problematic while covariance is not stable in time. While the method has been developed and tested based on spatially distributed soil moisture data, it is also suitable for analyzing other state variables.



## 1 Introduction

Hydrological systems are spatially variable and yet organized, which implies that a substantial part of the variability is not purely random but exhibits a systematic nature (Bronstert and Bardossy, 1999). Spatial organization can be described by the covariance of distributed observations. The covariance reflects spatially coherent fluctuations of the variable of interest

around a stationary mean. This implies that available observations bear predictive information about the variable at closely located points. The predictive information content declines to zero when the separation distance increases and approaches the correlation length. Spatial covariance is a key manifestation of spatial organization, because it reflects a configuration of a lower than maximum Shannon entropy (Shannon, 2001), as shown for example by Loritz et al. (2018). The strength of this form of structured variability of observations is usually characterized by means of a variogram and its characteristics (compare section

2). A spatial correlation structure in soil characteristics or rainfall implies also that within the correlation length hydrological processes experience similar controls. This correlation is a necessary condition for these processes to operate similarly. To account for this spatial organization, hydrological models for the saturated or the unsaturated zone often make use of either geostatistically interpolated or simulated parameter fields (Kitanidis and Vomvoris, 1983; Klaus and Zehe, 2010; Ly et al., 2011; Klaus and Zehe, 2011; Pool et al., 2015), initial states (Bronstert and Bardossy, 1999) or precipitation input (Goovaerts,

1999, 2000; Lloyd, 2005; Zehe et al., 2005; Haberlandt, 2007; Verworn and Haberlandt, 2011).

In principle, geostatistics relies on the idea of a largely time invariant spatial dependence of the field of interest (Burgess and Webster, 1980). Time-invariance is, however, rarely the case in hydrological systems. Non-orographic rainfall for example is highly variable in space and time. This implies that we can at best identify an average covariance structure of single rainfall events (Verworn and Haberlandt, 2011; Haberlandt, 2007; Zehe et al., 2005). In turn, rainfall and soil characteristics drive

soil moisture dynamics during and to some extent after rainfall events. During rain-free periods soil water dynamics are controlled by redistribution, soil evaporation, plant transpiration and capillary rise (Western et al., 1999; Seneviratne et al., 2010). The spatial covariance of the soil water content at a given time reflects the multitude of these influences and bears fingerprints of their respective covariance structures (Bárdossy and Lehmann, 1998). In consequence, the covariance structure of soil moisture cannot be stable in time. In fact, Western et al. (2004) found a seasonal evolution of the correlation length

in Australian catchments and attributed this to seasonal change in the processes controlling the soil moisture pattern. Schume et al. (2003) reported that the spatial continuity of water content is largely dependent on the drying and rewetting history of the soil. Grayson et al. (1997) stressed the existence of several preferred states in the soil moisture patterns in the Tarrawarra catchment and attributed those to different dominant controls. During wet conditions, Grayson et al. (1997) found lateral water movement through both surface and subsurface flow. Subsequently, wet areas were organized along these drainage lines. Thus,

spatial patterns in soil moisture are not time-invariant and should not be interpolated by means of geostatistics. In the light of this transient spatial dependence Lark (2012) concluded it as unlikely that the time-invariant covariance models used in soil geostatistics and soil hydrology can be as tightly linked to the processes as in certain areas of geophysics, such as gravimetric or magnetic fields. We therefore suggest that a thorough understanding of the temporal dynamics of the spatial dependence of soil



moisture observations will be helpful to decide when the application of geostatistical interpolation methods is feasible. This analysis can also indicate when the soil moisture patterns are dominated by meteorological and when by terrestrial controls.

The challenge of how to assess the time variant covariance and thus transient spatial similarity of soil moisture or rainfall has been addressed in several ways. The traditional way, to rely on a highly frequent and spatially dense spatial sampling of soil moisture sensors is restricted to the plot and hillslope scales (Brocca et al., 2007, 2009; Blume et al., 2009; Zehe et al., 2010). Interestingly, each of these studies reported the ranks of soil moisture observations to be fairly stable in time, despite the state dependent spatial variance. One alternative to frequent measurements is mobile and highly resolved soil moisture measurements during different meteorological conditions (Tyndale-Biscoe et al., 1998). This allows to cover larger extents, but at the cost of lower temporal resolution. Another approach is presented by Jost et al. (2009), who propose a stratified and nested sampling design, with 12 surveys over the course of 4 years. Here the locations are stratified along elevation, aspect and landuse zones to capture time variant patterns. In another study, using the same data, Jost et al. (2007) compare the shape and parameters of experimental variograms for the same sample at different points in time. Another different approach is the dense optimal approach to measure rainfall. This consists of a fairly uniform spatial distribution of the rain gauges, with an increased density in areas where the residual variance is larger (Wadoux et al., 2017). Taken together, these studies suggest different sampling strategies in order to target time variable patterns.

Here we follow a different avenue which relies on a context dependent analysis of soil moisture data in combination with time dependent variogram estimation. We are using a network of 45 sensor clusters, where each one measures soil moisture in three profiles at three different depths. We propose that soil moisture patterns during rainfall driven wetting conditions and during radiation driven drying essentially belong to different ensembles. The time series thus have to be classified based on their covariance and then the resulting classes need to be analyzed separately. More specifically we hypothesize that:

– H1: An ensemble of distributed soil moisture observations will converge to a rank stable configuration during long term drying periods, reflecting the time invariant spatial pattern of terrestrial controls. While rainfall events disturb this pattern, the soil moisture will re-organize to these stable ranks during drying.

– H2: Terrestrial controls, such as vegetation and soil properties, dominate the soil moisture covariance structure during long dry periods. This implies that the variogram of the soil moisture ranks converges against a stable configuration reflecting the covariance of the underlying time independent terrestrial controls.

## 2 The variogram as measure of similarity

As outlined, the spatial covariance structure of a distributed data set is frequently analyzed by means of the experimental variogram. More specifically by its well-known characteristics "nugget", "sill" and "range" (Burgess and Webster, 1980). Nugget, sill and range are usually estimated by fitting a theoretical variogram model (figure 1) to the experimental variogram. The nugget is the y-axis intercept of a variogram indicating the intrinsic variability in the data which is not explainable based on the spatial configuration of the observation points. The sill denotes theoretical maximum of the semi-variance, which implies





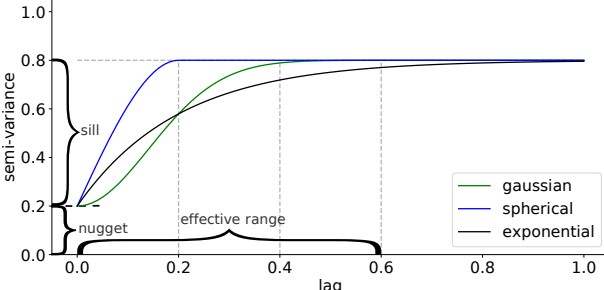

**Figure 1.** The three most common theoretical variogram models: spherical (blue), exponential (black) and Gaussian (green). All three variograms share the same variogram parameters: $nugget = 0.2$, $sill = 0.8$ and $range = 0.2$. Note that the range describes the range parameter $a$ in the variogram function, not the effective range $r$ shown in the figure, which is the length at which a spatial dependence is present. The latter is by definition the lag at which 95% of the sill are exceeded, because Gaussian and exponential models approach the sill asymptotically. Thus, they do not have a true range. For calculating the range parameter $a$, its relation to the effective range $r$ is defined as $a = 1 * r$ for the spherical, $a = \frac{1}{2} * r$ for the Gaussian and $a = \frac{1}{3} * r$ for the exponential variogram function. Thus, there are three vertical lines marking the effective range of the respective models.

that the semi-variances flatten out. The effective range is the corresponding separation distance where the variogram reaches the maximum, and corresponds to the separation distance at which observations become statistically independent. Hence, two identical variograms describe the same covariance structure.

As already stated we use the experimental variogram primarily to quantify whether and how spatial similarity of distributed
soil moisture observations evolve in time. We do not use the variogram for interpolation purposes. The latter requires that the intrinsic hypothesis is fulfilled, which is not the case if variogram structures are transient. The intrinsic hypothesis is weaker form of the second order stationarity hypothesis about a random variable. It requires the expected value of that variable and of all pairwise differences to be zero (Akin and Siemes, 1988). Note that our approach does also not require a fit of a theoretical model to the experimental variogram. The latter is necessary to calculate weights for kriging interpolation (Akin and Siemes,
1988).

The fundamental idea of the variogram is easiest illustrated by looking at theoretical model functions. In geostatistics, the most frequently applied variogram functions are the spherical, exponential and Gaussian model. An example of each is shown in Figure 1. The spherical and exponential functions have essentially a similar shape but their effective ranges are different. The low-range spherical variogram describes observables with spatial dependencies over short distances. Exponential variograms
describe correlation lengths over longer distances. A Gaussian model is of a fundamentally different shape describing a different spatial dependency. It implies that the covariance on small lags is very high and drops faster than in the other models relative to the respective effective range. Thus, a variogram bears information about the covariance field in its parameters as well as its shape.



During long term drying periods we expect the covariance structure of soil moisture pattern to gradually evolve to a static structure. Soil or vegetation characteristics controlling the soil moisture recession might be reflected by this static covariance structure. To quantify whether this kind of convergence does indeed emerge we need to quantify the similarity of variograms, and in this respect we rely on their shape rather than on their parameters. Such a similarity assessment based on the experimental

variogram can reveal similarities in variance (sill), spatial dependence (nugget to sill ratio), correlation length (range) and structural similarity i.e. the variogram shape. Here, we use the variogram to relate the semi-variance of soil moisture ranks to the separating distance of the corresponding point pairs. As long as we do not intend to interpolate, unlike the theoretical variogram function, monotony in this relation is not required.

In order to group similar variograms, we propose a variogram distance measure in combination with a density based cluster-
ing. This variogram distance measure is a vector distance between variograms as defined in equation (4) and further described in section 3.3. This combination allows also to detect and analyze the dynamics of spatial patterns and to define *whether and when* patterns with *distinctly similar* spatial structures emerge. We seek for long lasting clusters of similar variograms, that show a low intra-cluster variability. As long as these clusters can be represented by a variogram that describes a non-random spatial depency in the underlying covariance field, we refer to this as spatial *organization*. We then regard the emergence of

similar variograms to be a manifestation of stationarity of the processes controlling the drying of the soil. As further elaborated in the discussion section, these thoughts imply that kriging interpolation is only feasible if the variogram is stable in time.

In order to detect whether spatial organization emerges we need to isolate periods that belong to the same population. Loosely spoken this means that soil moisture dynamics are driven by similar "forces". Hence we seek to extract the longest possible recession periods which are only marginally disturbed by small rainfall events. These periods are particularly helpful

to visualize how rainfall alters the spatial organization of soil moisture dynamics during dry periods. It also visualizes how strongly the rain- or throughfall-induced patterns superimpose on the characteristic influences of the terrestrial controls. It has to be stated that large variability in the input will most likely not only be caused by the rainfall pattern. As the study site is mainly covered by forest (see section 3.1), the throughfall pattern will be governing the variability, as it is a lot more variable than open land precipitation (Keim et al., 2006; Tromp-van Meerveld and McDonnell, 2006). This said, whenever we describe

a rainfall event in the following, a rainfall event altered in its spatial structure by the canopy is meant.

## 3  Methods and data

### 3.1  Study area and soil moisture measurements

Our analyses are based on the CAOS data set (Zehe et al., 2014), which has been collected in the Attert experimental watershed since 2012. The Attert catchment is situated in western Luxembourg and Belgium (Figure 2). Mean monthly temperatures range

from $18°C$ in July to $0°C$ in January. Mean annual precipitation is approximately $850\,\mathrm{mm}$ (Pfister et al., 2000). The catchment covers three geological formations, Devonian schists of the Ardennes massif in the northwest, a mixture of Triassic sandy marls in the centre and a small area on Luxembourg Sandstone on the southern catchment border (Martinez-Carreras et al., 2012). The respective soils in the three areas are haplic Cambisols in the schist, different types of Stagnosols in the marls area and



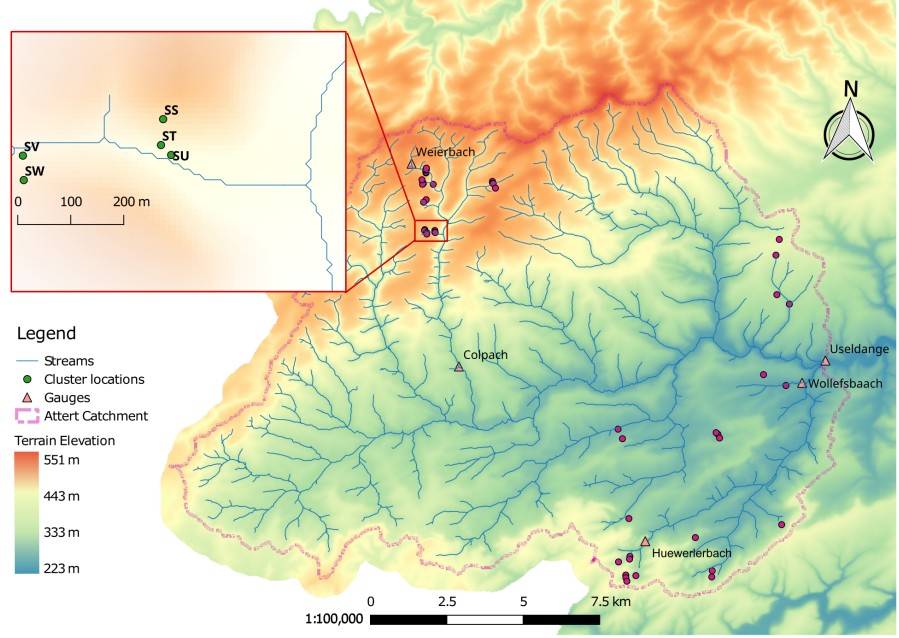

**Figure 2.** *Large Map:* The Attert catchment in Western Luxembourg and partly Belgium. The purple circles indicate the locations of the 45 sensor clusters. The underlying image is the terrain height. *Small Map:* The green circles mark the locations of the sensor clusters used in this study. All of them are located within the same small sub-catchment.

Arenosols in the sandstone (IUSS Working Group, 2006; Sprenger et al., 2016). The distinct differences in geology are also reflected in topography and land use. In the schist area, land use is mainly forest on steep slopes of the valleys which intersect plateaus that are used for agriculture and pastures. The marls area has very gentle slopes and is mainly used for pastures and agriculture, while the sandstone area is forested on steep topography.

5     The sampling design of the was intended to cover the variability of geology (schist, marls, sandstone), topography (different slope positions and aspect) and land use (deciduous forest and pasture) in the catchment. It consists of 45 sensor clusters which were installed in 2012-2013, their locations are indicated by the purple points in figure 2. At each cluster site, a range of meteorological and soil parameters have been measured; in this study we focus only on the soil moisture measurements (Decagon 5TE sensors). There are three soil moisture profiles at each cluster site, with soil moisture sensors at 10, 30 and 50

10 cm depth in each profile.

    As described in section 2, we focus on times of soil moisture recession. Three drying periods have been identified:

– **2013:** 01/06/2013 to 06/09/2013

– **2015:** 01/05/2015 to 15/08/2015

– **2016:** 15/07/2016 to 30/09/2016




The selection of these drying periods was done by visually selecting periods, where an overall decreasing trend could be identified over all sites in the Attert catchment. The data from 2014 has been excluded because there was no such undisturbed drying period during summer. In this study we seek for a subset of the CAOS data collected at comparable sites, while the objective during the CAOS data collection was to cover the landscape heterogeneity. The subset of cluster locations we used

in this study is shown in the smaller map in figure 2, indicated by the labeled green circles. The reasons for selecting this subset are explained in section 3.2 in more detail. The indicated subset covers five sensor clusters (SS, ST, SU, SV, SW) and will be further referred to as the *STU plot*, for brevity. In the following, we focus on the analysis of the data of the STU plot observed in the 2013 period. The same type of analysis for the other years and another plot (SA-SB-SC-SI) are provided in the supplementary material.

## 3.2 Experimental variogram estimation

To derive subsamples that satisfy the aforementioned requirements, we exclusively use observations from the same small scale sub-catchment or hillslope which have the same geology. The observation points are not separated by hilltops. The subset used for the presented analysis is the *STU plot* described in section 3.1 and shown in Figure 2. The data is aggregated to mean daily soil moisture values $\theta$. In line with our first hypothesis we did not analyze the soil moisture values, but their relative ranks.

As the direct observations are often skewed and subject to observation errors, these factors hamper the detection of spatial organization. In line with the study of Keim et al. (2005) , we expect that spatial organization manifests through rank stability. We thus transform the data into ranked values to make them more robust to outliers and signal noise. The ranks $R_t$ are identified for all $\theta$ at each time step $t$ with the highest $\theta$ receiving the highest rank. In order to increase the comparability between the different depths, the ranks are further transformed into a relative rank $P_t$ as described by equation (1). This way, two depths

with a different amount of sensors operating can be better compared.

$$P_t = \frac{R_t}{max(R_t)} \qquad (1)$$

where $max(R_t)$ is the highest rank value allocated at time step $t$. The relative rank represents the probability of an observation being smaller or equal to absolute soil moisture value at time step $t$. Stability of the relative ranks implies that the multivariate probability distribution of the soil moisture data is stable in space, despite the fact that the magnitudes of the individual

observations are changing.

Motivated and inspired by Chiverton et al. (2015), we estimate variograms from the value $z$ within a moving window defined by equation (2):

$$z(x_t) = \frac{\sum_{t=0}^{t+b} P_t}{b} \qquad (2)$$

for each time step $t = 1, 2, \ldots, (L - b)$ and a time series length of $L$ in days. The window size $b$ is here set to 5 (days). The

window size is further discussed in section 5.4. $P_t$ are the ranks given in equation (1); $x_t$ are the observation points that measured a value at time step $t$.





Each of these time windows is characterized by a variogram as an aggregating window function. The independent variable on the x-axis of the variogram is the separating distance lag $h$ between two observations. The maximum search distance for observation pairs was limited to the hillslope length of 240m. The lag classes are chosen to split the separating distances into six bins with the same amount of point pairs in each bin. The decision to use this binning over other procedures is further

discussed in section 5.4.

For all (observation) point pairs $(x, x + h)$ the semivariance $\gamma$ is calculated using the Matheron estimator (Matheron, 1963) defined by equation (3):

$$2\gamma(h) = \frac{1}{N_h} \sum_{i=1}^{N_h} \{z(x_i) - z(x_{i+h})\}^2 \tag{3}$$

where $N_h$ is the amount of point pairs at lag $h$ and $z(x)$ is described by equation 2 for every window location $t$. This methodol-

ogy is applied to all three periods in all three depths. Thus in total 9 result sets were obtained, further referred to as ensembles, yielding $N_h - b$ (up to 93) variograms per ensemble.

### 3.3   Variogram similarity

To quantify the similarity of variograms we use a density based clustering approach in combination with an appropriate distance function. Each variogram is understood as a vector in $\mathbb{R}^n$, where $n$ denotes the number of bins and is set to $n := 6$ throughout

this study. The Euclidean distance $d$ between two of these vectors is used as a measure of similarity as defined in equation (4).

$$d(v, w) = \sqrt{\sum_{i=1}^{n} (v_i - w_i)^2} \tag{4}$$

where $n$ is again the number of bins in both variograms $v, w$. A distance of 0 would in fact identify two completely identical variograms. The distance is well defined in our case, as we ensured that all bins are filled in all variograms. This would not be the case when using a constant lag step.

Furthermore, in the $\mathbb{R}^n$ space, a cluster of similar variograms is defined as a region of high variogram density. This density-based approach is different from a common approach of clustering multivariate data by fixing the number of clusters and just filling the members into the closest cluster. Fukunaga and Hostetler (1975) reported a *Mean Shift* algorithm, that perfectly serves our requirement for density-based clustering. It does not need the number of clusters as a parameter, but determines this number automatically, which is another requirement. Mean Shift is an algorithm, that can identify local maximum in a density

function (Cheng, 1995). It is an iterative approach that will shift points (in a 2D case) within a search radius (called bandwidth) onto the point closest to the highest density. The density is estimated by a kernel function, which is further explained below. The process is stopped only when all points are shifted and, therefore, only one point is left within the range of one bandwidth. These remaining points are denoted as cluster centroids, which are the points closest to highest density. The results of a Mean Shift is controlled by the shape of the chosen kernel and the bandwidth.



Mean Shift estimates the density function from a discrete data set. To do so, each point in $R^n$ is represented by a mathematical function called a kernel function, or kernel. On each iteration, the weighted mean of estimated density within the bandwidth is calculated. The distance of each point to this mean is used to shift the point. Thus, the kernel influences the weight of each point. We used the flat kernel (Cheng, 1995), which assigns the same weight to all points (equation (5)).

$$K(v_i) = \begin{cases} 1, & ||v|| \leq n \\ 0, & ||v|| > n \end{cases} \qquad (5)$$

where $v$ is either of the variograms $v, w$ used in equation (4) and $||v||$ is the vector norm of $v$, as it is understood as a vector at this stage as described above. The dimensionality of the search window is given by $n := 6$.

Identifying the appropriate bandwidth is important, as Mean Shift will fail to identify clusters which are closer together than the bandwidth, as these clusters will be shifted towards the point of highest joint density. In case the bandwidth is too small,
Mean Shift will identify sub-clusters, that would naturally be one single cluster. This will lead to lower inner-cluster variability than under large bandwidths (Comaniciu and Meer, 2002). Thus there is a trade-off between the number of clusters vs. inner cluster variance. By visual inspection the variograms seemed to frequently form three clusters. We therefore set the search distance to the 30% percentile of all pairwise Euclidean distances found within the population, because uniform distribution of variograms over the moving windows would then yield three clusters. Larger search radii would not be able to identify these
clusters and therefore the 30% percentile seemed to be a reasonable choice.

An example of the Mean Shift behavior in comparison to a very similar and common clustering algorithm called *k-Means* is shown in figure 3. k-Means is based on the distance between points and will sort them into a pre-defined number of clusters. In the upper row of figure 3, 80 points are randomly chosen from four normal distributions, each one located and scaled slightly differently. Both algorithms perform similarly, identifying more or less the same clusters. k-Means was set to four clusters.
Mean Shift also identified the four clusters. In the lower row of figure 3, we increased the sample size to 1500, drawn from the same distributions as before. While k-Means still identifies the same clusters, drawing artificial borders in the big cloud, Mean Shift is based on densities and identifies only two clusters. Neither is more correct, the choice depends on the aim of the study. The behavior of Mean Shift is the one we desire here as we would describe the last example by two clusters. More details and the actual implementation of Mean Shift we used can be found in Comaniciu and Meer (2002). In our application,
the variogram vectors can be constrained by the possible semivariance values. Following equation (3), the semivariance cannot be negative. Thus, all the variogram vectors have to lie in the positive quadrant of $\mathbb{R}^6$. The upper bound for the semivariance is the sill, by definition. With the chosen 6 bins, we end up with a maximum vector distance between two variograms of six times the overall variance for two nugget-effect variograms: $\gamma = 0$ and $\gamma = variance$, respectively.

Following Comaniciu and Meer (2002) we define the centroid to be the variogram with the smallest distance to the point of
highest density within the corresponding cluster. The centroid is thus a distinct variogram that was calculated for an specific time window and not a product of aggregation. The clusters may be regarded as soil moisture patterns with a distinct covariance structure and the centroids consequently are the variograms which characterize these patterns in a highly representative fashion.



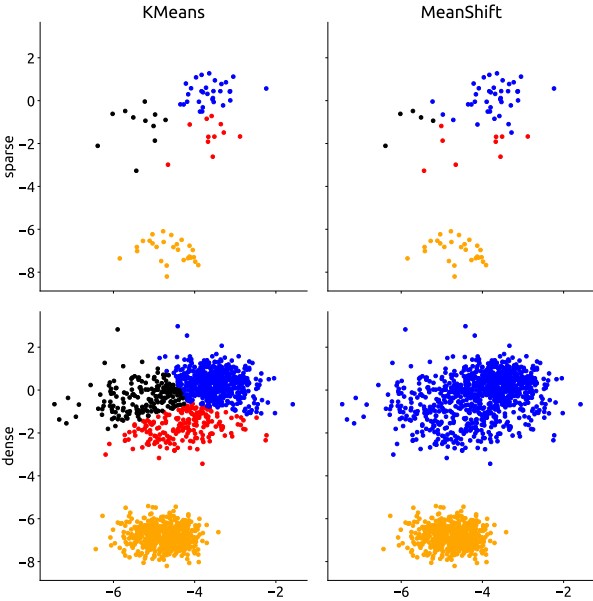

**Figure 3.** *Upper Row:* 80 random two-dimensional samples drawn from four normal distributions of different location and scale. The left column shows the result of a k-Means clustering with the color scheme indicating the different clusters. The right column shows the points clustered by a Mean Shift algorithm. *Bottom Row:* The same results as in the upper row based on a sample size increased to 1500.

The cluster centroid variograms will still be experimental variograms. In order to describe them by the variogram parameters effective range, sill and nugget an monotonous function is needed. Instead of fitting a theoretical variogram function to it, these variograms are monotonized. This procedure will turn the experimental into a robust estimation. The monotonized experimental variogram is, like the theoretical function, monotonously ascending, but does not imply any assumption about

5   the variogram shape. The conceptual procedure of monotonizing experimental variograms, their properties and a comparison to variogram functions are described in Hinterding (2003, section 4.4.2, p. 71f.). The actual implementation we used is called PAVA algorithm (Barlow et al., 1972). The three variogram parameters can be extracted from a monotonized variogram in the same way as from a theoretical variogram function. The nugget is the semivariance value at lag $h = 0$, the sill is 95% of the maximum semivariance value and the effective range the $h$ at which it occurs.

10   For each ensemble (i.e. the 9 combinations of periods and depths), the number of identified clusters gives an estimation of dominant states that can be found within the analyzed period and for the respective depth. The assignment of clusters could over time:

   – switch rapidly from one cluster to the other,

   – remain constant over longer periods,



- show a smooth gradual change in their inner cluster variabilities that spans over a long time and cannot be clustered properly.

A temporally persistent cluster clearly separated from other clusters would infer a high degree of persistent organization. This means the spatial structure described by the variograms is present over a long time and showing no or little variability. We assume this to happen during long, undisturbed drying periods. A gradual shift in variogram parameters would suggest the presence of only one process drying the soil, without setting a new spatial structure. Then, the smooth change in variogram parameters would reflect the interplay of the forcing pattern vanishing and the terrestrial pattern taking over.

The most important definitions of commonly used terms and their meaning in the scope of this work are given below:

- **Variogram similarity:** Let a variogram be a vector in $\mathbb{R}^n$ with $n := 6$, the number of lag classes, then the similarity between two variograms can be measured by their distance (see equation (4)).

- **Spatial structure** is the state of a covariance field of the observations. Here, we describe and quantify spatial structure by the variogram parameters and its shape.

- **Cluster centroid variogram:** It is the one variogram within a cluster, that is closest to estimated local variogram density.

- **Cluster variability:** As we describe a cluster of similar variograms by the properties of the centroid variogram, the variability in variogram parameter and shape within this cluster is a measure of representativeness of the centroid for the entire cluster.

- **Spatial organization:** Here, spatial organization is described by the number of clusters of similar variograms and their variability. Less inner-cluster variability is understood as higher spatial organization.

### 3.4 Uncertainty analysis

To assess the robustness of the obtained variogram clusters we analyzed the uncertainty propagating from the observations into the identification of clusters. To this end two main sources of uncertainty were considered: the measurement error caused by the sensor precision and the effect of complete sensor failures on the analysis, To properly propagate the resulting uncertainty through the analysis, we used a bootstrapping approach. Sensor failures are simulated by omitting a different randomly chosen sensor in every time step. The measurement error is simulated by adding uncorrelated noise with mean zero and a variance of $\sigma = 0.01$ to the data. Each type of bootstrapping contained 5000 repetitions to calculate the 95% confidence interval for both sources of uncertainty.

## 4 Results

### 4.1 Time dependent variogram similarity

The soil moisture measurements at 30 cm depth taken in the summer recession 2013 at the STU plot are shown in figure 4 as an example. Two kinds of minor changes in relative ranks can be observed: first, two neighboring sensors switch position




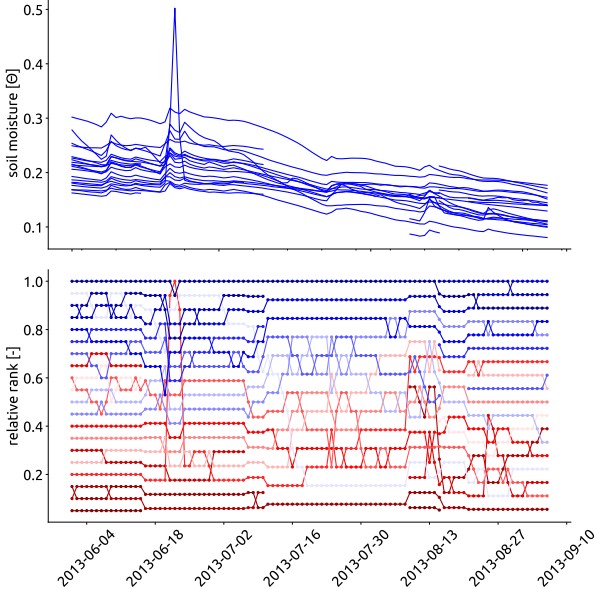

**Figure 4.** *Top:* Daily soil moisture observations from the 30 cm sensors used in this study. *Bottom:* The same data as in the top column transformed to relative ranks. The $y = 1.0$ line marks the highest observation for each day, while the smallest observation is plotted to the $y = 0.0$ line. The color of each line indicates the mean rank over the whole period, with the most saturated blue line holding the highest mean and the most saturated red line holding the smallest mean rank.

back and forth and second, a number of sensors change their relative rank value without any position switch. Back and forth switches might be subject to signal noise, while the gradual rank shift in the second case is caused by sensor failures, which will change the rank of the remaining sensors. A few sensors never stabilize in their rank positions during the entire period. But most sensors show rank stabilities lasting for weeks if not months. This is especially the case in the first month of the drying

period (Figure 4).

Rainfall input disturbs the spatial organization of the drying process by altering the ranks in soil moisture. This can be observed very clearly at the end of June, when one sensor is considerably more affected by the rainfall event than the others. This event does also show how spatially variable rainfall / throughfall input into the system superimposes a new pattern on the existing ranking. Similar observations can be made at the beginning of August, when the sensors show different responses to a

rainfall event. However, after the forcing ceases, the sensor rank reorganize themselves back into their intrinsic order (Figure 4). For a few sensors, long term shifts in ranks can be observed. These shift their relative rank position over periods of weeks and without rainfall input, while the majority is stable.

In case of a gradual change of the soil moisture pattern with time one would expect the variograms to change their shape gradually. This would be due to a gradual shift in covariance from the event based control (rainfall) to terrestrial controls (soil

and vegetation properties). In fact the variograms do change with time, but this change occurs step-wise or threshold-like and



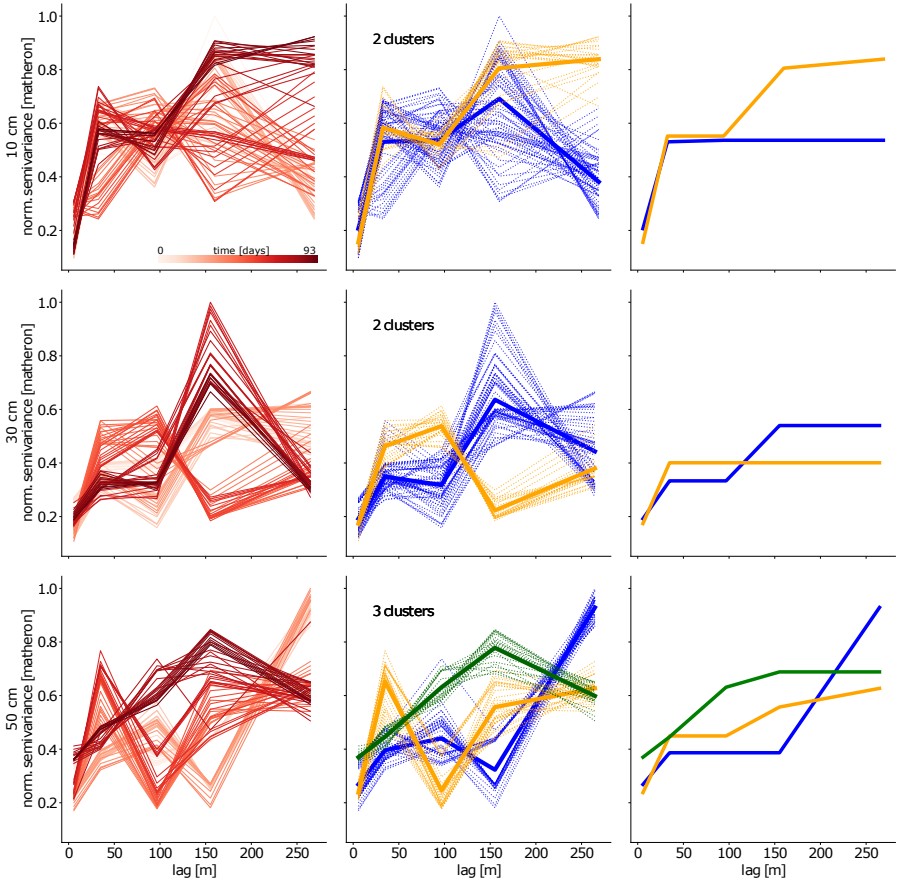

**Figure 5.** *Rows:* The rows represent the depth of the respective sensors with the first line being installed at 10 cm, the middle row at 30 cm and the bottom row at 50 cm depth. *Left:* The left column shows all experimental variograms resulting from a moving window over the drying period in 2013. The color indicates the window shift of that variogram with darker variograms occurring later in the period. The semivariance values, calculated from the normalized rank values, on the y-axis have been normalized to the maximum value within each ensemble (row). *Middle:* The middle column shows the same variograms from the left column as dashed lines. Here, the color indicates the clusters yielded by the Mean Shift algorithm. Solid lines represent the cluster centroid variograms. *Right:* The same cluster centroids already presented in the middle are shown in the right column. These cluster centroids have been monotonized and are thus monotonously ascending.

variograms seem to cluster temporally (Figure 5, left column). The threshold-like rapid changes of the variograms manifest through either a clear change in the range or variogram shape itself (Figure 5, left column). A clear shift in nugget could not be observed. It is also remarkable that the discrimination of clusters becomes clearer with increasing depths, as the gaps between the cluster members (Figure 5, left column, middle and lower plot) become bigger. This is consistent with the expectation that

5    the soil moisture pattern in shallow depths is much more sensitive to throughfall or precipitation disturbances than soil moisture in larger depths, because the overlying soil acts as a low-pass filter.



## 4.2 Clusters of variogram similarity

Following the clustering procedure defined in section 3.3, we clustered the variograms into similar groups (Figure 5, second column). Please note that the number of clusters is not predefined. We see that the number of clusters is increasing with depth and, more importantly, the variograms within each cluster become less variable. The cluster centroids indicated by the solid

line are actual variograms most representative for each cluster and not a product of aggregation (see section 3.3).

    The share of variance explained by the cluster centroids (Figure 5, right column) is expressed by the nugget/sill ratio. Interestingly, the nugget/sill ratios change over depth and among the clusters. Most ratios are between 0.2 and 0.4 and in both cases these ratio shifts are due to a change in the sill, while the nugget remains invariant. At the same time, the effective range changes. In 10 cm correlation lengths increase over time. In 30 cm the range is decreasing. In 50 cm there is a decrease

between the blue and orange and an increase in range between the orange and green cluster. However, there is a constant increase in effective range with time (as the variograms get more reddish in Figure 5, left column). The differences between states are controlled by change in variance (sill) and correlation length (range), while the micro variance close to the point of observation (nugget) stays the same. The sill is increasing over time. For the 30 cm depth (Figure 5, right column) the sill is dropping. However, in the left column all darker colored variograms show higher sill values. This cannot be reflected by the

cluster centroid as the aforementioned variograms are (mis-)classified into the blue cluster.

    Although no theoretical functions have been used, all the monotonized cluster centroids show a comparable shape. The shapes of the variograms for the monotonized centroids in 10 cm and 30 cm depth are similar to an exponential or a spherical function. At a depths of 50 cm the results are different. While the first (blue) variogram is similar to a Gaussian shaped theoretical variogram, the last one (green) looks most like a exponential one. The orange variogram, which has been observed

during the intermediate period also has an intermediate shape. To summarize, we find clusters that essentially differ in correlation length in 10 cm and 30 cm depth and clusters that change their variogram shape from a step-wise Gaussian structure to a continuous exponential structure in 50 cm depth. The latter could be caused by the absence (exponential) and presence (Gaussian) of groundwater, as further elaborated in the discussion.

    Looking at the temporal sequences in a bit more detail we find that at 10 cm depth variograms that initially oscillate between

different shapes at a period of a few days (Figure 6, left column). Later on, i.e. with increasing position on the y-axis, the variograms become more stable, and a stationary pattern can be observed for longer periods. Note that an increasing y-axis position represents an evolution in time, and thus overall dryer condition in the soil. Hence, the variograms shape become less variable with the progression of the soil moisture recession. In contrast, the variograms at 30 cm (middle) and 50 cm (right) depth appear already less variable in shape for longer periods at the beginning of the drying period. Especially the 50 cm depth

shows little varioagram changes over time within the three distinct clusters. In 30 cm the variogram shapes are less variable than in 50 cm, but more variable than in 10 cm depth. The variogram clusters persist over longer time periods in the deeper layers. The switch from one state to the other occurs rapidly, without any noticeable transit period.





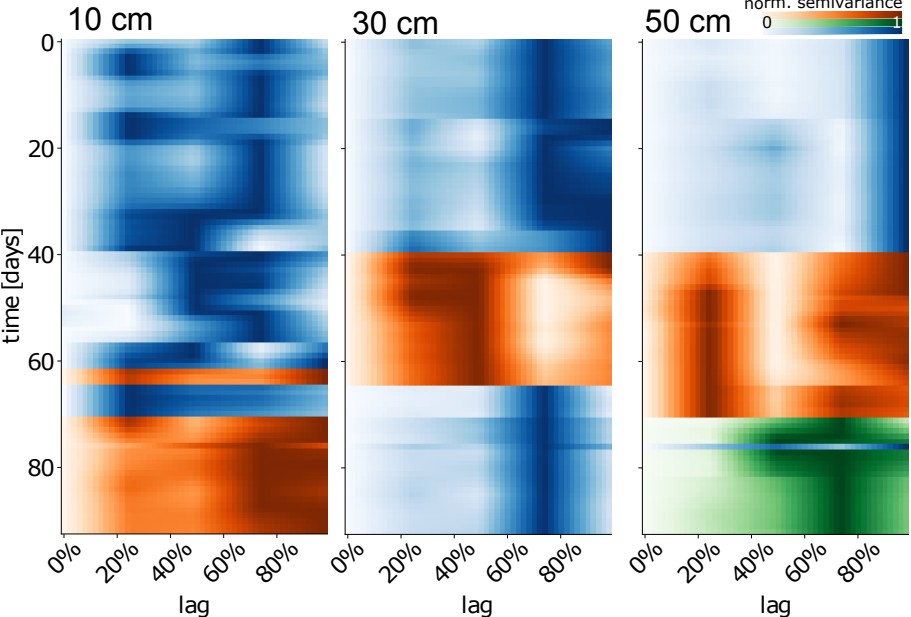

**Figure 6.** The columns represent the 2013 results from 10 cm (left), 30 cm (middle) and 50 cm (right). The moving window variograms are shown with the separating distance lags normalized to the maximum distance of 240m on the x-axis. The y-axis indicates the window position and therefore the temporal dimension. A number of e.g. 40 means, that the variogram in this row represents a window shift of 40 days from drying period start. The saturation of each pixel represents the semivariance at the given lag normalized to the maximum semivariance of the same variogram. The pixel color indicates the cluster the variogram has been grouped into. Pixels between the calculated lag classes have been interpolated by linear interpolation between the two neighboring lag classes, for visual reasons. Due to the normalization per variogram, the nugget value gets lost in this figure.

This analysis was also applied to the drying periods of the other years and to the other site. The resulting figures can be found in the supplementary material. The same kind of organization can be observed in most figures, in some cases somewhat less pronounced.

### 4.3   Uncertainty analysis

5   We analyzed the uncertainty propagation throughout the analysis using bootstrapping. The first step was to bootstrap the sensor precision. In a second step sensor failures were simulated and bootstrapped, as described in section 3.4. During bootstrapping, some of the simulations produced variograms which the Mean Shift algorithm failed to cluster into the same amount of clusters as presented in the results. Here, not only the variogram parameters of the cluster centroids changed, the whole variogram was dislocated and more or less cluster centroids were identified. These simulations can not be used for calculating the confidence

10   interval as they failed to capture the observed spatial dependencies. The share of these simulations has to be taken into account when looking at the the 95% confidence intervals as they add additional uncertainty not shown in the confidence intervals. On



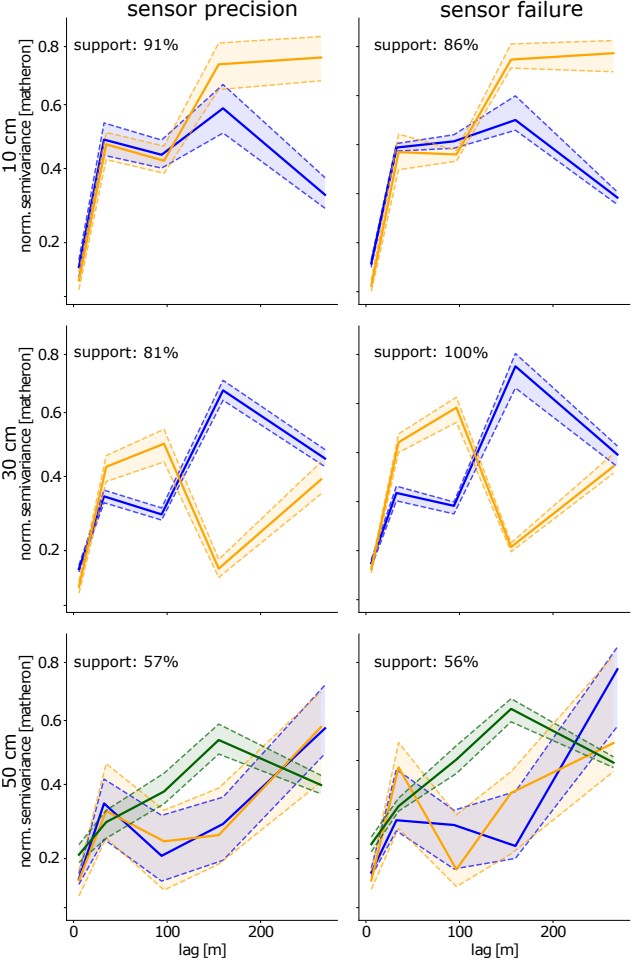

**Figure 7.** Result of the bootstrapping uncertainty analysis. The rows represent the different sensor depths at 10 cm (top), 30 cm (middle) and 50 cm (bottom). The colors of lines and areas represent the same clusters as shown in Figure 5, middle and right column. *Left column:* The cluster centroids identified by the Mean Shift algorithm are shown as solid lines surrounded by the 95% confidence intervals (dashed, filled area) calculated from the sensor precision simulation. *Right column:* The cluster centroids and 95% confidence intervals resulting from the sensor failure simulation. The sub-figures are aligned as in the left column.

the other hand, a high share of simulations ending up with the same cluster centroids can therefore be interpreted as measure of robustness. A low robustness implies that the identified cluster centroids could change dramatically, by changing minor amounts of data. The clusters identified for the 10 cm depth are supported by 91% of the simulations for sensor precision (Figure 7, upper left plot) and 86% for sensor failure (Figure 7, upper right plot). At 30 cm depth 81% of the sensor precision simulations identified the same clusters, while 100% of the sensor failure simulations matched the previously identified clusters (Figure 7, middle row). For 50 cm depth (Figure 7, lower row) only 57% and 56% of the bootstrap simulations matched the





three previously identified clusters. The other simulations generally identified only two clusters with a negligible amount of one and four cluster identifications.

For 10 cm and 30 cm depth neither sensor failure nor the assumed low sensor precision alters the variogram shapes substantially. The confidence interval is wider in 10 cm than in 30 cm. The sensor precision and sensor failure simulations show

comparable confidence intervals. In 50 cm, the green cluster is clearly distinct from the other two. The confidence intervals of the orange and blue cluster in fact suggest that these clusters could also be merged together. This is also stressed by the poor support of three cluster identification simulations, as two cluster identifications during bootstrapping are very likely.

In all presented bootstrapping results the confidence interval for the first bin is very narrow. Thus, the nugget did not change in any of the simulations. In 10 cm and 30 cm depth the whole confidence interval is very narrow and does not allow substantially

different variogram shapes. The sill will be subject to only small adjustments. Thus, the nugget/sill ratio can be considered to be robust against the tested sources of uncertainty.

## 5  Discussion

### 5.1  Hypothesis I: soil moisture reorganization

The main idea of this study is the use of time dependent variograms and variogram clustering to assess periods of meta stable

covariance structure during soil moisture recession periods. We emphasize that the variogram has not been used to estimate kriging weights.

In line with our first hypothesis, we showed that rainfall or throughfall events disturb the ranking among soil moisture measurements. During undisturbed drying periods, the ranks tends to re-organize into the same order that was observed prior to the disturbance. Similar findings of spatial patterns in soil moisture being disturbed by rainfall, but re-organizing after the

event are reported by Martini et al. (2015). This is quite interesting as it suggests that there is an immanent geogenic structure, which reflects the soil properties and gets superimposed by heterogeneous rainfall forcing. Our soil moisture observations were more often ranked stable in 30 cm and 50 cm depth than in 10 cm. This declining influence of external disturbance with depth can be explained by the more direct exposure to disturbances of the shallow soil and the fact that soil is a low pass filter. Martini et al. (2015) also found the spatial pattern in the topsoil to be more dynamic than in deeper layers. They identified preferential

flow to be the most influential factor for the formation of stable spatial patterns in these layers. This would imply that spatial patterns during storm periods would be more dependent on the coherent spatial pattern caused by preferential flow paths than on rainfall patterns.

It should be mentioned that our method was only applied in a limited scope. We would expect the distributed rainfall covariance field to be more arbitrary, changing from event to event, while covariance fields caused by throughfall and preferential

flow are likely to be more stable. However, in this study similar variograms were identified in the drying period on a single hillslope. This limited the amount of processes to be considered when analyzing spatial patterns in soil moisture. Thus, we cannot extrapolate rules of similarity to other time periods or spatial extent. Secondly, as also pointed out by Lark (2010): *"A shared variogram might mask substantial differences in the properties of the underlying random fields and of the processes*





*that they emulate."*; thus the variogram clusters must always be investigated in line with the ranks of the signal and its forcing. Otherwise, the identified similarities, structure or organization might be meaningless.

It has to be noted, that we are not the first to investigate time-variant spatial pattern. The coupling of variograms and a measure of temporal stability based on rank stability was proposed by Keim et al. (2005), who used it to study temporal
persistence in throughfall patterns. Fathizadeh et al. (2014) transferred the method from stand scale to the individual tree. Temporal persistence of spatial patterns using rank stability is widely used and published (Martínez-Fernández and Ceballos, 2003; Brocca et al., 2009). However, this is the first study that links the emergence of rank stability after re-organization to a single, representative variogram characterizing the covariance field being present during rank stability.

## 5.2   Hypothesis II: emergence of stable patterns

In line with our second hypothesis we indeed found emergence of meta stable patterns, which can be represented by a typical variogram. The time dependent variograms form distinct clusters. During drying periods we could observe a convergence towards stable structural configurations in spatial patterns, especially in deeper soil layers. However, this was not an entirely smooth convergence as changes between different meta stable patterns did not happen gradually, but rapidly (often within a single day). This supports the idea of sudden change in the dominant process forming the covariance field. Similar stable states
were previously found for soil moisture in a spatially dense monitoring network (Martini et al., 2015), but also for runoff formation (Western et al., 1998). We think that our results provide evidence that time dependent variograms in combination with density based clustering are a feasible means for detecting similarity over time and thus meta stable covariance patterns in the soil moisture ranks. The search for similarity in variograms and the question of how well they form clusters with distinct patterns over time can hence be addressed with the experimental variogram. Yet we monotonized the cluster centroids to
facilitate comparability. This way, the monotonized cluster centroids can represent the covariance of meta stable patterns by the variogram parameters effective range, sill and nugget.

We found our soil moisture measurements to be rank stable over longer periods (weeks). During these periods we could identify covariance structures that are well captured by a variogram. That means the cluster centroid variogram has a descriptive shape, a low nugget and is supported by most variograms in the cluster (low intra-cluster variability) and time-persistent
clusters of similar variograms can therefore be well described by their centroids. A combination of the cluster centroid and the intra-cluster variability may describe the period at which the cluster occurs better in terms of spatial dependencies than classic statistical moments. The number of clusters occurring and how well they represent the single variograms is then a measure of organization in the system. At the same time, as also set out in section 4.2, the number of clusters and their internal variability is not independent, it is a trade-off relationship. At smaller bandwidths Mean Shift will identify more clusters of
less internal variability. This internal variability will also decrease in case organization is happening in the soil over depth and over time. That said, a decreasing cluster variance should be investigated in the context of the applied clustering method. Thus, an intercomparison can only be meaningful in case the bandwidth parameter of Mean Shift is fixed. This might lead to some miss-classification as described in the results (section 4.2). But only then can the number of clusters be a meaningful measure of organization.





## 5.3 Application

Our proposed method can be used to detect spatial structures emerging from terrestrial controls and identify time-blocks with stable patterns. The identified cluster centroids could in principle be used for interpolation purposes, as they represent a meta stable covariance pattern. But this implies that maps as results of interpolation are only valid within distinct periods and that the

nature of the covariance may change rapidly within a few days. Interpolation during periods where controls on soil moisture dynamics shift from atmospheric to terrestrial influence is therefore not feasible as patterns are changing rapidly. Applied anyway, one would produce a map of two different things combined into one interpolation. Somewhat geogenic soil moisture pattern on the one hand and spatially distributed, heterogeneous forcing input carrying its own spatial structure on the other hand.

Once a robust estimate of a variogram shape is derived, the different shapes also reveal fundamental differences in the underlying covariance structure. The spherical and exponential variogram shape describe a gradual decay of the information content in the dependent variable over distance that converges to zero at the effective range. They differ in the gradient of this decay. A Gaussian variogram reveals a fundamentally different relationship that can be conceptualized to a step-wise function of a spatial structure being present or absent. Moreover, Gaussian semivariance value decays at low rates over short distances.

Up to a certain distance, the separating lag does not matter. All observations within this radius would receive essentially the same kriging weights. The cluster centroids in figure 5, 10 cm (first row) and 30 cm (middle row) show either spherical or exponential shapes, while the 50 cm (last row) also show different shapes. All three clusters together, can neither be fitted to a Gaussian, nor to an exponential variogram function. The covariance in 50 cm depth is best described by a Gaussian function at the beginning of the drying period. This suggests that fundamentally different kinds of disturbances are present at this depth.

These are characterized by high correlations on short distances (see section 2). However, at the end of the period, the covariance is best described by an exponential function again, just like at the other depths.

## 5.4 Limitations and challenges

It needs to be stressed that a variogram shape is highly dependent on the binning procedure. This is true for both experimental variograms and fitted theoretical variogram functions. Therefore a theoretical variogram function might mask the influence of

binning, at the cost of increased uncertainty.

In this study, the binning was implemented to always satisfy the requirement for the same amount of point pairs in each bin when defining the lag classes. We chose this approach because the variogram is used as input vector to the Mean Shift clustering algorithm and therefore must not hold any empty bins which would result in an undefined vector. At the same time it ensures robust variogram estimation where each semivariance calculation is based on a constant sample size, but it comes

at the cost of bins that differ in width. Some bins represent observations with more or less the same separating distance range. Others collect observations from a wider range of separating distances, that are compressed to the bin midpoint.

Another point of consideration is the window size for the moving variograms which we set to five days. We chose this period as we expect direct response of soil moisture to rainfall events to take place at smaller temporal scales within the context of this





site (of medium and high topographic gradients). A change to 4 or 6 days of window size did not change the results notably. Much larger windows would aggregate too much of the drying period into one variogram. Nevertheless, we cannot derive any insights beyond the temporal resolution of the chosen window. Short term rapid dynamics will by design not be detected by the presented method.

Lastly, this study relies on the fundamental assumption that proximity in separating distances can explain covariance in the observed data. This is an assumptions immanent to all geostatistical studies. Nevertheless, separating distances expressed as Euclidean distances of Cartesian coordinates might still not be capable to reflect the true proximity of observations. Therefore it might be enlightening to change the independent variable of the variogram to a measure that can better incorporate hillslope-scale heterogeneity and base a similar study on this measure. A good basis for such distance measure could be a topographic

index (Moore and Thompson, 1996) or an incorporation of surface gradient indices (Pérez-Peña et al., 2009; McCleary et al., 2011).

## 6   Conclusions

The presented diagnostic approach based on estimating variograms with a moving temporal window (Chiverton et al., 2015) proved useful for detecting, separating and describing metastable spatial states of organization in soil moisture observations.

These metastable states were not only found in the presented data, but also for other years and sampling plots presented in the supplementary material. We conclude that structural organization of soil moisture has preferred states with distinct patterns, dependent on their respective drivers, rather than fluctuation between a large range of possible structural realizations.

We think that the presented method is particularly useful to improve kriging workflows. Although the derived cluster centroids cannot (yet) be used for calculating kriging weights, they can be used to identify a temporally stable spatial configuration

and the time period over which it occurs, where kriging is feasible. Further research should focus on the incorporation of cluster centroids into existing kriging procedures.

*Code availability.* The analysis is based on the scikit-gstat Python package. The exact version used is 0.1.8 (Mälicke and Schneider, 2018). The analysis scripts can be found at https://gitlab.com/mmaelicke/moving-variograms, and will be made public as soon as this work has been published.



*Acknowledgements.* We thank the German Ministerium für Wissenschaft, Forschung und Kunst, Baden-Württemberg for funding the V-FOR-WaTer project. We thank the German Research Foundation (DFG) for funding of the CAOS research unit FOR 1598. We especially acknowledge Britta Kattenstroth and Tobias Vetter, the technicians in charge of the maintenance of the monitoring network. Special thanks go to Andràs Bàrdossy for the enlightning discussion, that paved the way for this work. The authors also acknowledge support by Deutsche

5  Forschungsgemeinschaft and the Open Access Publishing Fund of Karlsruhe Institute of Technology (KIT). The service charges for this open access publication have been covered by a Research Centre of the Helmholtz Association.





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
