# Peer review of "Exploring hydrological similarity during soil moisture recession periods using time dependent variograms"

_Hydrology and Earth System Sciences, 2018_

## Referee Comment (RC1) · Anonymous Referee #1 · 12 Oct 2018

This is an interesting study, but there are some failures of clarity, and ultimately I am unconvinced by the analysis.

The authors raise the general point that the spatial covariance parameters of soil water content (SWC) are unlikely to remain fixed as the soil wets and dries. This is a reasonable point. I think that the authors' treatment of it would gain considerably in clarity if they expressed it in terms of current practice for space-time geostatistical modelling. They state from time to time that one cannot interpolate SWC data because of the possibility that covariance parameters change with time, but this is not true if one interpolates for a single time, and if one interpolates from a full space-time sample then

one requires a space-time model, not just a spatial model even when one can be defined. Various space-time models are available, of which the simplest are the separable models. But even a relatively simple non-separable model, the product-sum model, requires that we can assume a marginal spatial variogram. The authors' contention is that such a marginal variogram cannot be defined for SWC because of changes in the spatial parameters over time.

I am not convinced by the approach that the authors have taken. In summary they want to cluster empirical spatial variograms to identify time periods in which these appear to be stationary. But it is not clear that this is useful. If one's objective is to interpolate within a period corresponding to a cluster one is still ignoring the temporal variation and the temporal dependence that it might exhibit. It would be more appropriate to develop an extension of existing space-time models, using appropriate methods to deal with non-stationarity (Pintore, A., Holmes, C.C., 2005. A dimension-reducing approach for spectral tempering using empirical orthogonal functions. In: Leuangthong, O., Deutsch, C.V. (Eds.), Geostatistics Banff 2004. Springer, Dordrecht, pp. 1007–1015; Sampson, P.D., Guttorp, P., 1992. Nonparametric-estimation of nonstationary spatial covariance structure. Journal of the American Statistical Association 87, 108–119).

Second, I do not think that ranks are an appropriate way to tackle this problem. Rank statistics throw away a lot of information, and their statistical distribution makes them unsuited to most geostatistical modelling.

Finally, it is surely clear that a network of 15 sensors is entirely inadequate to estimate the parameters of a spatial model. Aggregating over time might give the illusion of respectable sample sizes, but given the temporal dependence which is to be expected for SWC this could be seriously misleading.

In summary, I think that there is the germ of an interesting study here, but it requires a more adequate data set, and appropriate models based on a stronger conceptual

approach as might be offered by space-time geostatistical models.

---

## Author Comment (AC1) · 18 Oct 2018

Dear Referee,

Thanks for your helpful comment on our work. Your comment is highly appreciated.

First of all, we have to clarify that the objective of the presented study is not interpolation. We did not want to give the impression that geostatistics are generally not suitable. Space-time geostatistics can surely be used to interpolate based on spatial as well as temporal dependencies in the data.

Additionally we wanted to stress that we think that during drying and wetting different

processes are dominant. Thus data should be stratified according to their dynamic controls, and we think that the covariance reflects different controls.

In the revised manuscript we will clarify that a key focus of our study is on the memory of the soil. How long does it take until the soil 'forgets' about the rainfall disturbance in a sense that soil water dynamics and its spatial covariance are dominated by the soil and vegetation characteristics? This time scale will certainly depend on the observation depths (as the soil is a low pass filter), and the soil hydraulic properties (the water capacity and hydraulic conductivity). We think that the proposed approach is well suited to quantify this relaxation time scale.

From your comment we gather that our reasoning so far is not sufficient to motivate our approach. We will gladly take up your suggestions and will compare our method to the one you suggested to make it easier for the reader to understand the differences in the methods and the application of ours. In our study we want to demonstrate how long it takes for the dominant processes to switch and in what manner this transition happens. The evolution of a spatial dependency and how it converges into stationarity was found to be useful in this context. However, the studies named by the referee seem to be helpful and we are happy to test a replacement of the used variograms by the named space-time variograms as a moving window function. We will re-structure our introduction and methods to better clarify this main focus of our study.

Next, we completely agree that rank statistics take out a lot of information. This will surely complicate a geostatistical interpolation and the transformation back to absolute values. However, we are looking at a more general development and evolution of co-variance, or variogram shapes, in the recession of soil moisture. Here, the ranks might be more appropriate for exactly this decoupling from the absolute values. For comparison in our specific case, we re-run the moving variogram analysis part for the absolute values (figure 1). We used the Cressie estimator here, which is more robust on extreme values, just like the ranks are (Cressie, N., & Hawkins, D. M. (1980). Robust estimation

of the variogram: I. Journal of the International Association for Mathematical Geology, 12(2), 115–125. https://doi.org/10.1007/BF01035243). The variogram shapes are comparable and our study focuses on the clustering and shape of variograms. We would argue to keep the variograms based on ranks to be more consistent with the analysis concerning our hypothesis 1, which is based on ranks as well. We will clarify in the manuscript, that these variograms are not suitable for interpolation for the named reasons.

Finally, we agree that 15 sensors are not enough to estimate a proper spatial model that can be used for interpolation. In fact there is more data available, all located within the same geology. The other sensor locations can be found in figure 1 of the manuscript. Our original intention in limiting the sample size was to conduct a study on similarity evolution on the scale of one hillslope. We will test our method applied to more sensors and compare to the current results. This will, however, move the focus of the study to stable patterns developing on a much larger spatial scale.

In summary, we want to thank the referee for this most helpful comment on our work and hope that the outlined changes and clarifications will substantially improve the manuscript.
* * *
[Figure]

**Fig. 1.** Left: Result as in the study, based on semivariance of normalized ranks in 10 cm (top row), 30 cm (middle) and 50 cm (bottom); Right: Result based on Cressie-semivariance of absolute values

---

## Referee Comment (RC2) · Anonymous Referee #2 · 18 Nov 2018

The comment was uploaded in the form of a supplement: https://www.hydrol-earth-syst-sci-discuss.net/hess-2018-396/hess-2018-396-RC2-supplement.zip

---

## Author Comment (AC2) · 27 Nov 2018

**Response to Referee Comment on "Exploring hydrological similarity during soil moisture recession periods using time dependent variograms"**

by Mälicke et al.

November 2018

**Response**

Dear Referee,

Thanks for your helpful comment on our work. Your comment is highly appreciated.

We will revise section 3.3 in terms of comprehensibility. Technical terms referring to to the clustering method will be checked and explained.

We will improve the description of the study site and the corresponding map (manuscript, Figure 1). Additionally, we might include more locations into our analysis, please refer to our response to Referee #1 in this context.

As suggested by the referee, we tested different common clustering algorithms and ran the clustering on the actual data. The resulting comparison presented below and how we justify the selection of *MeanShift* will be added to the manuscript's method section. Note that a variogram is viewed as a coordinate tuple in $\mathbb{R}^6$ and therefore cannot be plotted easily. We rather visualized the classification result of four different clustering algorithms (Figure 1). This shows that *MeanShift* and *KMeans* result in the same classification. The main downside of *KMeans* is that the number of clusters has to be predefined, which makes it unsuitable for the study. *DBSCAN* is, like *MeanShift*, a density based algorithm. It failed to find more than one cluster in all investigated cases, most likely because our variograms are of rather high dimensionality ($\mathbb{R}^6$) compared to the sample size of 92 (variograms). *Affinity Propagation* sounded promising as it is based on the idea of finding the set of $n$ most representative instances within the sample. The *Affinity Propagation* is mainly controlled by the *damping factor*, which influences the calculation of similarity between two instances. Like the bandwidth parameter for *MeanShift*, different *damping factors* can result in just one or as many clusters as sample size. However, we preferred the

[Figure]

Figure 1: Clustering result for the 30cm variograms presented in the manuscript using four different algorithms: *MeanShift*, *KMeans*, *DBSCAN* and *Affinity Propagation*. The time lag of the moving window is shown on the x-axis and the cluster membership of each variogram is indicated by the color. Same color means same classified cluster.

*MeanShift* as the bandwidth is directly linked to the Euclidean distance between two coordinates (variograms) in the sample and could therefore be inferred from the variogram distribution. The *damping factor*, on the other hand, seems more like a dimensionless calibration factor tweaking the results and therefore it would be way more complicated to select an overall justifiable value.

Regarding the physical connection of the manuscript's Figure 5, we will consider adding either a rainfall or throughfall time series to the manuscript. This could be done in Figure 4, as it would align with the soil moisture observations and their ranks over time. Including a rainfall time series per sensor, as suggested in the minor comments, will not be possible, as rainfall was not observed at this high spatial resolution (note that some sensors are only a few meters apart). From our understanding the sudden changes in Figure 5 are either due to rain- /throughfall or a change to terrestrial controls dominating the described spatial structure. The presented sensor locations share similar vegetation and soil characteristics, therefore investigating their influence does not seem to be promising for the scope of the current study. A soil map of the current study site would locate them all in the same soil unit. However, as more locations might be included into the revised study, this is an important and interesting aspect for discussion and will be considered.

While we cannot discuss 2015 and 2016 in as much detail as 2013, we will add a brief methodological assessment covering these two years to the revised manuscript.

Concerning the other minor comments:

- *p.5 l.9:* The sequence of the equations will be revised.

- *p.6 l.5:* The sentence will be reworded to be grammatically correct.

- *p.7 l.1-5:* The whole section 3.1 will be revised, as more sensors will be included into the study. Special attention will be given to the marked paragraph.

- *p.8 l.20:* We mean similar in terms of variogram shape. This will be clarified.

- *p.8 l: 25-30:* This paragraph will be revised and, as indicated above, technical terms will be defined.

- *p.12, caption of Figure 4:* As described above, a rainfall time series per sensor is not possible, but rainfall data will be added to the study.

We would like to thank the referee for these helpful comments and will improve our manuscript by connecting our results better to other physical observations in the field, along with a substantially improved and clarified description of the clustering method.